# Red Blood Cell-Derived Microparticles Exert No Cancer Promoting Effects on Colorectal Cancer Cells In Vitro

**DOI:** 10.3390/ijms23169323

**Published:** 2022-08-18

**Authors:** Dania Fischer, Fabian Thies, Omar Awad, Camilla Brat, Patrick Meybohm, Patrick C. Baer, Markus M. Müller, Anja Urbschat, Thorsten J. Maier, Kai Zacharowski, Jessica Roos

**Affiliations:** 1Department of Anesthesiology, Intensive Care Medicine and Pain Therapy, University Hospital Frankfurt, Goethe University, 60590 Frankfurt am Main, Germany; 2Department of Anesthesiology, Heidelberg University Hospital, 69120 Heidelberg, Germany; 3Department of Safety of Medicinal Products and Medical Devices, Paul-Ehrlich-Institute (Federal Institute for Vaccines and Biomedicines), 63225 Langen, Germany; 4Department of Anesthesiology, University Hospital Wuerzburg, 97080 Wuerzburg, Germany; 5Clinic of Internal Medicine III, Division of Nephrology, University Hospital Frankfurt, 60590 Frankfurt am Main, Germany; 6German Red Cross Blood Transfusion Service and Goethe University Clinics, 60590 Frankfurt am Main, Germany; 7Institute for Transfusion Medicine and Immunohematology, 34117 Kassel, Germany; 8Clinic of Urology and Pediatric Urology, Philipps University of Marburg, 35043 Marburg, Germany

**Keywords:** transfusion, red blood cells, microparticles, colorectal carcinoma

## Abstract

The biomedical consequences of allogeneic blood transfusions and the possible pathomechanisms of transfusion-related morbidity and mortality are still not entirely understood. In retrospective studies, allogeneic transfusion was associated with increased rates of cancer recurrence, metastasis and death in patients with colorectal cancer. However, correlation does not imply causation. The purpose of this study was to elucidate this empirical observation further in order to address insecurity among patients and clinicians. We focused on the in vitro effect of microparticles derived from red blood cell units (RMPs). We incubated different colon carcinoma cells with RMPs and analyzed their effects on growth, invasion, migration and tumor marker expression. Furthermore, effects on Wnt, Akt and ERK signaling were explored. Our results show RMPs do not seem to affect functional and phenotypic characteristics of different colon carcinoma cells and did not induce or inhibit Wnt, Akt or ERK signaling, albeit in cell culture models lacking tumor microenvironment. Allogeneic blood transfusions are associated with poor prognosis, but RMPs do not seem to convey tumor-enhancing effects. Most likely, the circumstances that necessitate the transfusion, such as preoperative anemia, tumor stage, perioperative blood loss and extension of surgery, take center stage.

## 1. Introduction

Red blood cell (RBC) transfusion has been associated with poorer prognosis in oncologic patients, notably in patients with colorectal cancer, but also in patient cohorts with other malignancies [1,2,3,4,5,6,7,8]. Although primarily based on observational studies with their intrinsic limitation to adjust for known and unknown confounders, reports on potential adverse effects of RBC transfusions became a great source of insecurity among patients and clinicians. Randomized trials to explore whether the relation between blood transfusion and poor prognosis is causal or coincidental are difficult to undertake due to the ethical impossibility of actively over or undertransfuse patients in light of hemotherapy guidelines on the one hand and manifold potential confounders on the other hand. The major bias persists that patients with more advanced disease stages, severe surgical trauma or higher amounts of blood loss might receive more transfusions.

Few animal studies were undertaken to address the issue in a setting with fewer confounders. Atzil and colleagues, for instance, found that the transfusion of RBC units is an independent and significant risk factor for cancer progression in murine tumor models of mammary adenocarcinoma and leukaemia [9]. They tried to identify whether red cells, leukocytes or soluble factors from RBC units display cancer-promoting effects and found that aged erythrocytes, rather than leukocytes or soluble factors, were accountable for the effects in a storage time- and cell volume-dependent manner. Lin and colleagues compared the effects of allogeneic vs. syngeneic vs. autologous RBC transfusions in a murine model of a solid sarcoma [10]. They found that allogeneic blood transfusion was associated with an increased rate of tumor growth and postulated that this was mediated by major histocompatibility complex incompatible antigens, incurring immunomodulatory effects in the transfusion recipients.

Multiple factors in RBC products have been suggested to play a potential role in cancer progression, including apoptotic cells, cytokines, extracellular vesicles, free hemoglobin and iron [11,12,13,14]. We were interested in the role of RBC-derived microparticles (RMPs) as they carry abundant bioactive molecules, including different forms of nucleic acids and proteins, that can markedly modulate cellular behavior and eventually tumor progression by promoting phenotypic modification and reprogramming of cell functions [12,15]. Generally, microparticles are nanosized extracellular vesicles, which are generated from a broad spectrum of cells and comprise a very heterogeneous group of particles, containing various proteins, lipids, DNA, mRNA and noncoding RNAs [16]. RMPs are shed routinely from RBCs during normal maturation in vivo; yet this process is accelerated during processing and storage for transfusion. Of note, RMP composition differs from that of intact RBCs, and the nature and composition of RMP components are affected by both storage duration, the character of storage solutions and the method of isolation [17].

Recognized RMP bioactivities include promotion of coagulation, immune modulation and the promotion of endothelial adhesion, as well as influence upon vasoregulation [18,19,20,21].

We therefore hypothesize that, firstly, RMPs might have a biological effect on cancer cells and that this effect might differ depending on the method of isolation. Secondly, we hypothesize that RMPs affect functional and phenotypic characteristics of colorectal cancer cells.

The primary aim of this study was to investigate whether RMPs influence growth, invasion, migration and tumor marker expression in colorectal cancer cells in vitro, and whether this depends on the method of RMP isolation. We were furthermore interested in potential RMP-effects on the WnT signaling pathway, which demonstrates hyperactivation in the majority of colorectal cancers and is perceived as the initiating and driving event of tumor development [22,23]. We therefore tested two RMP isolation protocols and conducted an in vitro study to determine whether RMPs would affect the growth, invasion, migration and tumor marker expression of HCT-116, HT-29 and Caco-2 colon carcinoma cells and WnT signaling in HEK293T cells.

## 2. Results

In this study, RMPs from 36 different RBC units were isolated, counted via flow cytometry (FC), and used for cell stimulation. On average, the donors were 42 years old (median 43). A total of 31 RBC units were isolated by ultracentrifugation (UC), as the most common isolation method, and five units by size exclusion chromatography (SEC) (for an isolation scheme see Appendix A). We chose SEC as a second isolation method since several reports demonstrated that the structural characteristics and biological effects of microvesicles could be influenced by the method used to isolate MPs [24]. Flow cytometric identification of RMPs was aided by Megamix-Plus SSC Beads in order to allow for a standardized and reproducible quantification. Figure 1A,B show the gating strategy.

RMPs were analyzed for phosphatidylserine exposition by Annexin V staining and the RBC marker CD235a. RMPs were counted with 0.3 μm sized PE fluorescent counting beads (Figure 1C–E). Depending on the RBC unit used as a source for RMPs, both the concentration of RMPs as well as the ratio between Annexin V/CD235a double positive to CD235a positive MPs varied greatly (Appendix A). This result was similar for both isolation methods. The RMPs were concentrated in a range between 5.3 × 10^2^ and 7.8 × 10^5^ RMPs/µL (mean of UC isolated RMPs: 1.3 × 10^5^ ± 2.0 × 10^5^ RMPs/µL; mean of SEC isolated RMPs: 8.6 × 10^4^ ± 5.2 × 10^4^ RMPS/µL). The rate of double positive MPs (Annexin V/CD235a to CD235a positive) differed between 15–98% (for UC and SEC), although in the majority of isolates (*n* = 19 out of 36), over 70% of CD235a positive MPs were also Annexin V positive. This distribution does not follow any obvious pattern, indicating a donor specific surface marker composition of the analyzed MPs, which is also in line with previous publications [25,26]. Notably, we could not observe any significant difference between the quantities of total RMPs isolated by either UC or SEC (mean 2.0 × 10^8^ ± 3.3 × 10^8^ RMPs (UC)/1.6 mL vs. 1.7 × 10^8^ ± 1.0 × 10^8^ RMPs (SEC)/2 mL; see Appendix A). In addition to the RMP concentration, we also determined the iron content of the RMP isolates since iron is also suspected to contribute to cancer development [27]. As illustrated in Appendix A only small traces of iron (<27 µ/dL) were detected in the majority of the analyzed RMPs (7 out of 11 isolates) and only four isolates displayed iron concentrations above 27 µg/dL (33–44 µg/dL). Interestingly, the isolation method seems not to significantly influence the iron quantity of the analyzed RMP isolates. Since FC does not allow for absolute measurement of MP size, we further analyzed the RMP isolates via nanoparticle tracking analysis (NTA). NTA allows determining the size of all particles with diameters of approximately 10–1000 nm, including microvesicles but also lipoproteins or protein complexes [28]. NTA measurements revealed that the mean size of the analyzed MPs was 198.6 ± 27.1 nm (Figure 1F), which lies in between the reported RMP sizes of 100–300 nm [28]. A direct comparison of the RMP concentrations measured via FC or NTA demonstrated much higher particle numbers for NTA than for FC analysis, emphasizing the higher specificity of the FC quantification method for MP detection (Appendix A, FC mean 1.4 × 10^4^ ± 1.7 × 10^4^ RMPs/µL vs. NTA mean 9.8 × 10^8^ ± 6.1 × 10^8^ particle/µL).

Next, we wanted to analyze whether RMPs isolated by the two different methods have different biological effects. We found that the incubation of HCT-116-cells with 10 × 10^6^ RMPs/mL and 50 × 10^6^ RMPs/mL only showed significant effects on cell viability when RMPs were isolated via ultracentrifugation (Figure 2). Cell viability rose to 117.3 ± 4.4%, *p* ≤ 0.05, and 143.7 ± 7.3%, *p* < 0.001, respectively.

Incubation with RMPs isolated via SEC did not reach significance, although the cell viability of HCT-116 cells increased to a similar extent when treated with 10 × 10^6^ RMPs/mL (108.2 ± 12.4%) and 50 × 10^6^ RMPs/mL (138.8 ± 10.4%). These results indicate that the effect of RMPs on tumor cell growth does not seem to be impacted by the isolation method used. To exclude cell-specific effects, we analyzed the effect of RMP on the viability of two further colon carcinoma cell lines, HT-29 and Caco-2. As can be seen in Figure 2B,C, increasing concentrations of RMPs, whether isolated via UC or SEC, did not affect the viability of HT-29 but reduced the viability of Caco-2 cells by approximately 30%. This effect could already be observed with 0.5 × 10^6^ RMPs/mL and did not further increase with higher RMP concentrations [mean Caco-2 for 0.5 × 10^6^ RMPs/mL: 78.4 ± 11.4%]. Since ultracentrifugation is the most common method for MP isolation, we decided to use only UC-isolated RMPs for follow-up studies.

To investigate possible long term effects of RMPs on proliferation of HCT-116 cells, a cell culture assay based on CFSE staining was performed over an extended period. We could not observe any significant effect on day two or day four when compared to PBS (median CFSE intensity day 2: 4919.1 ± 1435.5 vs. 6038.0 ± 1818.7; *p* = 0.095, median CFSE intensity day 4: 600.8 ± 176.6 vs. 899.3 ± 279.3; *p* = 0.078 (Figure 3A–D).

As PCNA (proliferating cell nuclear antigen) is a proliferation marker, associated with tumor proliferation and invasiveness in colorectal carcinoma, we aimed at detecting changes in protein synthesis [29]. Incubation for 24 h with RMPs did not result in changed synthesis of PCNA (mean of 5 × 10^6^ RMPs/mL: 102.3 ± 27.0% vs. mean of 10 × 10^6^ RMPs/mL: 108.7 ± 36.4%) (Figure 3E,F). Migration and invasion also play an important role in tumor progression. Therefore, we used an in vitro model where HCT116 was tempted to migrate and invade in a transwell chamber with or without extracellular matrix. We used FCS as a chemoattractant. Incubation with RMPs only resulted in significantly greater migration of HCT-116 cells compared to the control when 20% FCS was in the well (2.8 × 10^4^ ± 3.3 × 10^3^ relative fluorescence units [RFU] versus 1.8 × 10^4^ ± 5.0 × 10^3^ RFU, *p* = 0.006) (Figure 4A).

None of the other assay conditions tested had a significant effect on cell migration. Regarding invasion, the incubation with RMPs resulted in significantly greater invasion of HCT-116 cells compared to control when the FCS content in the well was 0%, and thus there was no FCS gradient (3.8 × 10^3^ ± 9.9 × 10^2^ RFU vs. 2.4 × 10^3^ ± 4.6 × 10^2^ RFU, *p* = 0.02) (Figure 4B). None of the other assay conditions tested rendered RMPs to have a significant effect on cell invasion.

Because the Wnt signaling pathway is already aberrantly activated in HCT-116 cells, we performed the reporter gene assay in HEK293T cells and found that RMP incubation did not inhibit or activate the pathway (Figure 5 and Appendix A). Additionally, we also analyzed β-catenin and GSK-3β in HCT-116 cells, which mediate increased proliferation, migration, and invasion via activation of the Wnt/β-catenin pathway in colorectal carcinoma. However, RMP incubation affected neither localization, expression of β-catenin and GSK-3β nor GSK3-β phosphorylation of (whole cell lysates: mean of 5 × 10^6^ RMPs/mL = 85.7 ± 17.1% (β-catenin), 78.7 ± 12.5% (p-GSK-3β), 117.5 ± 23.1% (GSK-3β) vs. mean of 10 × 10^6^ RMPs/mL = 74.7 ± 24.2 (β-catenin), 86.5 ± 15.9% (p-GSK-3β), 105.4 ± 24.4% (GSK-3β)). Next to the Wnt pathway, we further analyzed the effect of RMPs on the activation of ERK and Akt signaling via Western blot analysis of whole cell lysates, cytosolic and nuclear fractions. Yet, RMPs did not significantly inhibit or activate either pathways [(Figure 6 and Appendix A), whole cell lysates: mean of 5 × 10^6^ RMPs/mL = 115.4 ± 23.95% (p-Akt), 99.4 ± 12.6% (Akt), 109.3 ± 53.1% (p-ERK-1/2), 108.6 ± 16.7% (ERK-1/2) vs. mean of 10 × 10^6^ RMPs/mL = 126.7 ± 14.2% (p-Akt), 102.7 ± 10.0% (Akt), 115.7 ± 49.7% (p-ERK-1/2), 125.2 ± 11.4% (ERK-1/2)].

## 3. Discussion

The biomedical consequences of allogeneic blood transfusions and the possible pathomechanisms of transfusion-related morbidity and mortality are still not entirely understood [30]. In retrospective studies, the transfusion of RBCs was associated with increased rates of cancer recurrence, metastasis and death in patients with colorectal cancer but also in patient cohorts with other malignancies [1,2,3,4,5,6,7,8]. However, as correlation does not imply causation, these empirical observations have to be elucidated further in order to reduce insecurity among patients and clinicians. There are many confounding factors, such as preoperative anemia, severity of illness, perioperative blood loss, variations in hemotherapy algorithms and extent of surgical trauma, that affect the outcome and complicate clinical studies [31,32]. Furthermore, it is unethical to withhold RBC transfusions when indicated, which is why there are no prospective clinical studies comparing RBC transfusion vs. no transfusion. However, it is known from observational studies that the preoperative reduction of anemia and avoidance of RBC transfusion is associated with a better mid-term surgical oncologic outcome, although the effect is probably multimodal [33].

The purpose of this in vitro study was to elucidate the abovementioned empirical observation in regard to the oncological outcome after RBC transfusion in cell culture models, as we are convinced that it is important both for patients and clinicians to know the spectrum of potential treatment side effects. We focused on the in vitro effect of RMPs that are contained in RBC units. Our own previous research demonstrated that RMPs affect various biological processes such as hemostasis and immune response, which is why we hypothesized that RMPs might also contribute to the pathogenesis of colorectal cancer [18]. This has already been shown for MPs from other cell lines, such as platelets. Platelet-derived MPs play a significant role in tumor angiogenesis and they can furthermore act as a direct tumor growth enhancer through the release of potent growth factors in the tumor micro-environment [34,35,36]. As most retrospective studies on the effect of allogeneic transfusion on oncological outcome were undertaken in colorectal cancer patients, we incubated HCT-116 cells with RMPs and analyzed their effect on growth, invasion, migration and tumor marker expression.

However, methodologically, it has to be stated that the content of MPs is highly dependent on the circumstances mediating microvesiculation, and the characteristics of MPs studied ex vivo highly depend on the methodology of isolation. The RBC units have been shown to contain a mixed population of MPs, their concentration, composition, as well as their effects on the quality of the blood product vary depending on the manufacturing methods used to produce the RBC unit, which might also include blood donor characteristics such as age and gender [12]. As there are many scientific reports that plasma proteins change with age, i.e., anabolic mediators such as insulin-like-growth factor, it might also be possible that RMPs change with donor age and, hence, have different effects on cancer cells. We used 36 different RBC units as a source for RMPs to include a wide diversity of blood donors and isolated the RMPs on day 42 of storage so that storage duration was the same in all units [37]. Nevertheless, it might be worth testing whether RMPs change with donor age or gender.

The surface antigens of MPs allow for the identification of their cellular origin as well as their functional characterization. From our previous work, we know that the RMP quantity in RBC units rises during storage [18]. It is assumed that an average RBC unit contains 2.0 × 10^8^ RMPs on storage day 42, but presumably, significant losses and MP denaturation are associated with MP isolation and, hence, the isolates do not exactly mirror the MP content in the original source [38]. Mol and colleagues found differences in the protein synthesis of endothelial cells after stimulation with MPs isolated via SEC compared to UC-MPs [39]. Our results from comparing both methods imply that SEC works for RMP isolation as numerical RMP yield was comparable to UC but that surface antigen expression and iron content vary (Appendix A). Despite the differences, cell stimulation with RMPs isolated with UC and with SEC both resulted in comparable cell viability. Since we used RMPs from many different blood donors and different isolates with different proportions of CD235a/Annexin V positive vesicles in the assays (marked red in Appendix A) and tended to observe the same effect, we do not presume that the method of isolation makes a difference, at least in the assays we used. As the effect on tumor growth presumably does not depend on the method of RMP isolation, we did not further investigate different methods of isolation and moved on with RMPs isolated via UC.

We analyzed a concentration-dependent effect and found conflicting effects on cell viability. We observed an increase in cell viability at very high RMP-concentrations in HCT-116 cells (Figure 2). In contrast, we observed a reduction in viability in the Caco-2 cells, whereas there was no effect in HT-29 cells. However, since the effect observed in Caco-2 cells is not concentration-dependent and also not dependent on the isolation method, we assume that it is a cell-specific or artificial effect. As the increase in HCT-116 cells only occurs after incubation with very high dosages, we do not assume clinical significance.

However, with regards to quantity, it has to be stated that MP size generally renders them difficult to quantify due to multiple technical pitfalls [40]. The most commonly used quantification method is via flow cytometry supported by Megamix-Plus SSC beads and antibody staining. As NTA cannot differentiate between MPs and other small particles such as lipoproteins or other protein complexes, it was predictable that NTA would count more events than flow cytometry, which is why we decided to use NTA only as a Appendix A. Our results are in line with other publications on systematic differences in the methods of quantification [41]. There is no ideal method of quantification, but as we quantified all isolates to be used for stimulation in the cell culture with flow cytometry, at least the same systematic error should apply to our in vitro results. As the initially used supra-high concentrations will never be clinically relevant, we limited the dosage in follow-up assays to 5 × 10^6^ RMPs/mL, which roughly correlates to a 440 mL/kg RBC transfusion (provided that RMPs remain biologically active and are not cleared from the circulation, which is unlikely). Although we used leukoreduced RBC units as a source for MPs, we found some MPs that were Annexin V^+^/CD235a^−^, which shows that MPs from other cell lineages were present as well (Appendix A). As we did not observe an effect on the cell lines, the contamination with MPs from other lineages, which might also affect tumor growth, seems negligible, making further investigations unnecessary. Regarding cell migration (Figure 4), effects were only seen when incubated without or at higher concentrations of FCS. Nevertheless, there might be a trend towards a small effect of RMPs both on migration as well as on invasion, although these conditions are very unnatural and components in serum might interact with the RMPs. Hence, further experiments are necessary to determine the potential role of RMPs in migration and metastasis.

As the Wnt signaling pathway is believed to be the initiating and driving mitogenic event in almost all colorectal cancers, we also wanted to observe whether RMPs might trigger changes in this signal cascade [22]. Because the Wnt signaling pathway is already aberrantly activated in different colon carcinoma cells, we performed the reporter gene assay in HEK293T cells and found that RMP incubation did not inhibit or activate the pathway (Figure 5). We found that the expression of GSK-3, the final step for the transcription of Wnt target genes and also a protagonist in further signaling pathways such as NFkB and P53/p21, was not affected by RMPs. Markers of tumor proliferation were also not increased by co-culture with RMPs in HCT-116 cells. Additionally, we studied Akt and ERK signaling, important in proliferation, cell cycle regulating and apoptosis in human colon cancer cells, and found no effect of RMPs in HCT-116 cells [42].

In summary, our results show that RMPs do not seem to significantly affect the functional and phenotypic characteristics of HCT-116 cells in vitro. The clinical impact of RMPs in transfusion recipients is an area of emerging investigation. Even though our laboratory observations can provide some potentially valuable information about the effects of RMPs, the (patho-)physiological relevance of RMPs in vivo remains unknown and cannot be determined on the basis of preclinical data alone. The main limitation of our study is that we used isolated cell culture models and supra-normal concentrations of RMPs. MPs from other cell types were identified as key players in pathophysiological responses, for instance by supporting angiogenesis [43]. To draw further conclusions on the role of RMPs, a test system more complex and closer to the situation in vivo rather than a single cancer cell culture model would be necessary. In particular, potential immunomodulatory effects of RBC supernatants and RMPs among various immunologic cell types remain an important area for future research.

Nevertheless, our results stand in contrast to the results of the retrospective studies that associate allogeneic RBC transfusion with a worse oncological outcome in colorectal cancer patients. Our study is just one small part of the puzzle, and there are many gaps to be filled regarding our understanding of the biomedical consequences of allogeneic blood transfusions and possible pathomechanisms of transfusion-related morbidity and mortality. In particular, colorectal cancer resection is characterized by numerous processes that induce an abrupt elevation in the risk of the outbreak of pre-existing micrometastases and the seeding of a new metastasis [44,45].

## 4. Materials and Methods

### 4.1. Cell Lines and Culture Conditions

HCT-116, HT-29, Caco-2 and HEK293T cells were obtained from the German Resource Centre for Biological Material. L-cells and L-Wnt3a cells were obtained from the American Type Culture Collection. HCT-116, HT-29, Caco-2 and HEK293T cells were cultured in Dulbecco’s Modified Eagle Medium (DMEM; Thermo Fisher Scientific, Waltham, MA, USA) supplemented with 10% or 20% (Caco-2) fetal calf serum (FCS; Thermo Fisher Scientific, Waltham, MA, USA), 100 U/mL penicillin, 100 µg/mL streptomycin (Sigma-Aldrich, St. Louis, MO, USA). L-cells and L-Wnt-3a cells were cultured in DMEM with 10% FCS, 1% non-essential amino acid solution and 1% sodium pyruvate (Sigma-Aldrich, St. Louis, MO, USA). L-Wnt-3a cell medium was also supplemented with 0.4 mg/mL G-418 (Capricorn Scientific, Ebsdorfergrund, Germany). All cell lines were regularly tested for mycoplasma contamination.

For the cell culture assays, HCT-116 cells were plated in DMEM without phenol red supplemented with 10% microvesicle depleted FCS, 100 U/mL penicillin, 100 µg/mL streptomycin and 2 mM L-glutamine (Thermo Fisher Scientific, Waltham, MA, USA).

### 4.2. Isolation of Red Blood Cell-Derived Microparticles (RMPs)

Human RMPs were isolated from red blood cell (RBC) concentrates obtained from the German Red Cross Blood Donor Service Baden-Wuerttemberg—Hessen (Frankfurt, Germany) on day 42 of storage as the quantity of RMPs increases with storage duration [17]. Appendix A shows the schemes for RMP isolation and processing. If not stated otherwise, all work steps were done at 4 °C. The RBC concentrate was centrifuged at 500× *g* for 15 min, the supernatant containing the RMPs was transferred into new test tubes, and the pelleted RBCs were discarded. To remove the remaining cell debris, the supernatant was centrifuged again at 4600× *g* for 20 min. After these two centrifugation steps, the supernatant was either subjected to ultracentrifugation (UC) or size exclusion chromatography (SEC). After the first ultracentrifugation step at 100,000× *g* for 90 min, the pellet was resuspended in phosphate buffered saline (PBS, 0.22 µm filtered) and ultracentrifuged again at 100,000× *g* for 90 min. The RMP containing pellet was resuspended in PBS (0.22 µM filtered), quantified and aliquoted before storage (−80 °C). For RMP isolation via SEC, the supernatant was concentrated to a volume of 0.5–2 mL using Amicon-Ultra/PLBC Ultracel-PL Membran centrifugal devices with a 3 kDa cut-off (4000× *g* for 40 min at room temperature; Merck Millipore, Burlington, MA, USA) [39,46]. The SEC column was prepared by packing a poly-prep chromatography column (Bio-Rad, Hercules, CA, USA) with 10 mL sepharose CL-2B (GE Healthcare, Chicago, IL, USA). After a washing step with 20 mL PBS (0.22 µm filtered), the column was loaded with the concentrated supernatant and fraction collection started immediately using filtered PBS as an elution buffer. The separated supernatant was collected in 13 sequential fractions, with the first fraction measuring 1.5 mL, and all the following fractions measuring 1 mL. This distribution was chosen based on preliminary experiments showing that RMPs were mainly concentrated in fractions 2 and 3 and most proteins and iron were eluted in later fractions (see Appendix A). Thus, we chose the combined fractions 2 and 3 for follow-up experiments. The quantification of RMPs was done by flow cytometry as described below. The protein and iron amount in each fraction were quantified using the Pierce™ BCA (bicinchoninic acid) protein assay kit (Thermo Fisher Scientific, Waltham, MA, USA) and QuantiChrom™ Iron Assay Kit (BioAssays Systems, Hayward, CA, USA) according to the manufacturer’s instructions. The SEC was conducted at room temperature, and the collected fractions were put on ice immediately. All fractions were quantified and aliquoted before storage (−80 °C). The isolated RMPs (UC and SEC) were thawed only once.

### 4.3. Quantification of RMPs by Flow Cytometry

Samples were analyzed using a FACS Canto II flow cytometer (BD Bioscience, Franklin Lakes, NJ, USA) with accompanying Diva software. Flow cytometry performance quality control was achieved by BD FACSDiva™ CS&T Beads. Data were elaborated using FCS Express^®^4 (De Novo Softwares, Boulder, CO, USA). Identical parameters were applied to all samples. All work steps were done at 4 °C if not stated otherwise. To adjust the flow cytometer settings for standardized MP analysis, Megamix-Plus SSC beads (Stago, Asnières-sur-Seine, France) were used according to the manufacturer’s instructions. Samples (30 µL per analysis, 1:30 diluted in PBS) were incubated with anti-CD235a-BV421, as a marker for erythrocytic membranes and Annexin-V as a marker for apoptosis and microparticles (MPs) in Annexin-V Binding buffer (0.22 µm filtered; all reagents from BD Bioscience, Franklin Lakes, NJ, USA; for detailed antibody description see Appendix A) for 30 min at room temperature. RMPs were defined as CD235a and Annexin V positive. Prior to staining, both reagents were filtered using Ultrafree-MC/Durapore-PVDF centrifugal 0.22 µm filter units (12,000× *g* for 4 min; Merck Millipore; Burlington, MA, USA) to eliminate false positive events as described by Inglis et al., 20,153 [47]. For RMP quantification, MP-Count Beads (30 µL per analysis; Stago, Asnières-sur-Seine, France) with a known concentration were spiked into each RMP sample. Samples were analyzed at low speed for 2 min. Lysed (0.5% NP-40) double stained samples were used to define the background fluorescence (negative control).

### 4.4. Nanoparticle Tracking Analysis (NTA)

Particle amount and size distribution of isolated particles (10 µL per analysis, 1:500 dilution in filtered PBS) were measured using a NanoSight NS500 instrument (Malvern Instruments Ltd., Malvern, UK). Of each sample, six videos of 30 s duration (at 25 frames per second) were captured with a camera level set at 12 and a monitoring temperature of 28 °C. The detection threshold was set to 16, and blur and Max Jump Distance were set to auto. Data was analyzed with the Nano Sight NTA software version 3.2 (Malvern Instruments, Malvern, UK).

### 4.5. WST-1 Cell Viability Assay

A total of 1 × 10^4^ cells/mL (HCT-116, HT-29, Caco-2) were plated in triplicate in 100 µL microvesicle depleted DMEM on a 96-well plate. After 24 h, the medium was renewed, and the cells were incubated with increasing concentrations of RMPs (0.5–50 × 10^6^ RMPs/mL) or vehicle control. After 48 h, the medium was changed and cell viability was assessed using WST-1 reagent (Sigma-Aldrich, St. Louis, MO, USA) according to the manufacturer’s instructions.

### 4.6. Migration and Invasion Assays

Assays were performed in 24-well plates with 8 µm membrane Transwell^®^ cell culture inserts (Corning Inc., Corning, NY, USA). For the invasion assay, the Transwell^®^ insert membrane was coated with 300 µg/mL Matrigel^®^ basement membrane matrix (BD Bioscience, Franklin Lakes, NJ, USA) for two hours at 37 °C, resulting in a thin layer of extracellular matrix above the membrane. HCT-116 cells were starved in serum-free DMEM for 24 h. After starving, 1 × 10^5^ cells were placed into the Transwell^®^ inserts. 10% and/or 20% FCS in microvesicle depleted DMEM were used as chemoattractant and were added into the receivers. 5 × 10^6^ RMPs/mL or vehicle control were either placed into the Transwell^®^ inserts or receivers. After 24 h, medium was removed from the inserts and receivers and both were washed twice with PBS. Migrated cells from the underside of the insert membrane were simultaneously detached and stained with a calcein acetomethyl (calcein AM; Biotrend GmbH, Köln, Germany) and cell dissociation solution (CDS; Trevigen, Gaithersburg, MD, USA). After 60 min incubation with the calcein AM/CDS solution, the inserts were removed and the fluorescence was measured using a Tecan Infinite M200pro plate reader (excitation/emission [Ex/Em] = 485 nm/520 nm; Tecan Group, Männedorf; Switzerland). Each measurement was performed in triplicate.

### 4.7. CFSE (Carboxyfluorescein Succinimidyl Ester) Cell Proliferation Assay

HCT-116 cells were stained with 5 µM CFSE (Thermo Fisher, Waltham, MA, USA) for 20 min at 37 °C (protected from light) according to manufacturer’s instructions. After staining (day 0), cells were washed with PBS, plated into 24-well plates (5 × 10^4^ cells/mL/well) and incubated with 5 × 10^6^ RMPs/mL or vehicle control for two and four days. On day 2 and 4, cells were harvested with Accutase (Sigma-Aldrich, St. Louis, MO, USA), fixed with 2% paraformaldehyde (20 min, room temperature, Sigma-Aldrich, St. Louis, MO, USA) and analysed using a FACS Canto II flow cytometer (BD Bioscience, Franklin Lakes, NJ, USA). Unstained cells were used as a negative control. Stained cells from day 0 were analysed to confirm uniform cell labelling. In the case of a not uniformly labelled cell population, unstained cells were excluded in the later performed analysis.

### 4.8. Western Blot Analysis

3.75 × 10^5^ HCT-116 cells/mL were plated into 6-well plates. On the next day, cells were incubated with two concentrations of RMPs (5 × 10^6^ or 10 × 10^6^ RMPs/mL), vehicle control, L-cell conditioned medium (LCM; no Wnt-3A protein; negative control), Wnt-3A conditioned medium (WCM; containing active Wnt-3A protein) for 24 h as well as EGF (50 ng/mL) for 30 min. After the respective incubation times, cells were harvested, either lysed on ice with cell lysis buffer (1.5 mL 1 M Tris HCl, pH 6.8, 3 mL of 20% SDS, 3 mL Glycerol, 19.5 mL H_2_O) followed by sonication or lysed with the nuclear extraction kit (Cayman Chemicals, Ann Harbor, MI, USA) according to manufacturer’s instructions. The protein amount was quantified using the Pierce™ BCA (bicinchoninic acid) protein assay kit (Thermo Fisher Scientific, Waltham, MA, USA). Equal amounts of proteins (50 µg) were separated by SDS-PAGE (SDS-polyacrylamide gel electrophoresis, 4–20% precast gel [BioRad, Hercules, CA, USA], or 12% Tris-Glycine gels) and separated proteins were electrophoretically blotted onto nitrocellulose membranes (Hybond-C Extra, Amersham Biosciences Ltd., Amersham, Pl Little Chalfont Buckinghamshire, UK). The membranes were blocked for 1 h at room temperature (Blocking buffer: Intercept blocking buffer [LI-COR Biosciences, Lincoln, ME, USA, diluted 1:1 with TBS, for immunoblot with PCNA] or 5% BSA in TBS), incubated with primary antibodies overnight at 4 °C and then incubated with secondary antibodies IRDye680^®^ or IRDye800^®^ (LI-COR Biosciences, Lincoln, ME, USA) for 1 h at room temperature. Antibodies were diluted in blocking buffer in the presence of 0.01% Tween-20. For a detailed antibody description see Appendix A. The Western blots were visualized using an Odyssey CLx infrared imaging system. After each protein detection, membranes were carefully stripped with 1× NewBlot nitrocellulose stripping buffer (LI-COR Biosciences, Lincoln, NE, USA).

### 4.9. Wnt Reportergene Assay

HEK293T cells were seeded into 96-well plates (2 × 10^4^ cells/well) and after 24 h, cells were co-transfected with either pGL4.49 (TOP; firefly luciferase) or pGL4.27 (FOP; firefly luciferase; negative control) and with pGL4.74 (renilla luciferase, transfection control) by the calcium phosphate precipitation method. After 24 h since transfection, cells were incubated with LCM, WCM or WCM plus one of two concentrations of RMPs (5 × 10^6^ or 10 × 10^6^ RMPs/mL). After 24 h, the luciferase activity was assessed via the Dual-Glo^®^ Luciferase assay system (Promega, Madison, WI, USA) using a Tecan Infinite M200pro plate reader (Tecan Group, Männedorf, Switzerland). The firefly signal was normalized to the renilla activity and to untreated cells. Each measurement was performed in triplicate.

### 4.10. Statistical Analysis

GraphPad^®^ version 5.0 (GraphPad Software, Inc., San Diego, CA, USA) was used for the statistical analysis. The dataset was tested for normal distribution using the Kolmogrov–Smirnov normality test. Statistical significance was either determined by an unpaired *t*-test (normal distribution), a Mann–Whitney analysis (no normal distribution), or a Kruskal–Wallis test followed by a Dunn’s multiple comparison test. The tests used for analysis are indicated in the figure legends.

## 5. Conclusions

In conclusion, allogeneic blood transfusions are associated with poor prognosis, but RMPs do not seem to convey tumor-enhancing effects. Certainly, enhancing our understanding of the role of hemotherapy in tumorigenesis and tumor development is vital for clinical perioperative practice. Most likely, the circumstances that necessitate the transfusion, such as preoperative anemia, tumor stage, perioperative blood loss and extension of surgery, take center stage when it comes to oncological outcome. Nevertheless, known and suspected side effects of hemotherapy alongside the limitations of the underlying evidence should be kept in mind. The indication to transfuse should be weighed carefully and follow current evidence-based guidelines.

## Figures and Tables

**Figure 1 ijms-23-09323-f001:**
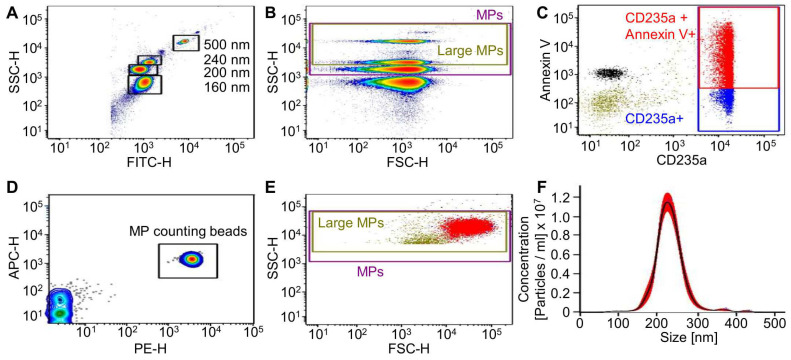
Quantification of red blood cell-derived microparticles (RMP). (**A**–**E**) Flow cytometer (FC) settings and gating strategy for RMP quantification with Megamix-Plus SSC and MP-Counting Beads (0.3 µm sized PE fluorescent beads). Shown are density/dot blots from one representative measurement out of 32 independent analyses. SSC-H vs. FITC-H (**A**) and SSC-H vs. FSC-H (**B**) density blot showing the location of the Megamix-Plus SSC beads. Beads were used to set up the SSC-H, FITC-H and FSC-H PMT voltage (**A**,**B**), the MP gates (**B**) and the SSC threshold for RMP analysis. (**C**) Gating strategy to characterize the RMP population. Dual APC/Annexin V vs. BV450/CD235a scatter dot showing the double stained RMP isolate. RMPs are defined as CD235a and Annexin V positive. (**D**) Dual-APC-H vs. -PE-H density blot showing the location of the MP-Count Beads during sample aquisition. Beads with a known concentration were spiked into each RMP sample. Samples were analyzed at low speed for at least 2 min or to a corresponding minimum of 500 MP-Count Beads. (**E**) SSC-H vs. FSC-H dot blot illustrating the double stained RMP isolate within the prior defined (see (**B**)) MP gates. (**F**) Nanoparticle tracking analysis (NTA) of RMP isolates via Nanosight NS500 (*n* = 4). Shown is one representative NTA out of four independent analyses of RMP isolates.

**Figure 2 ijms-23-09323-f002:**
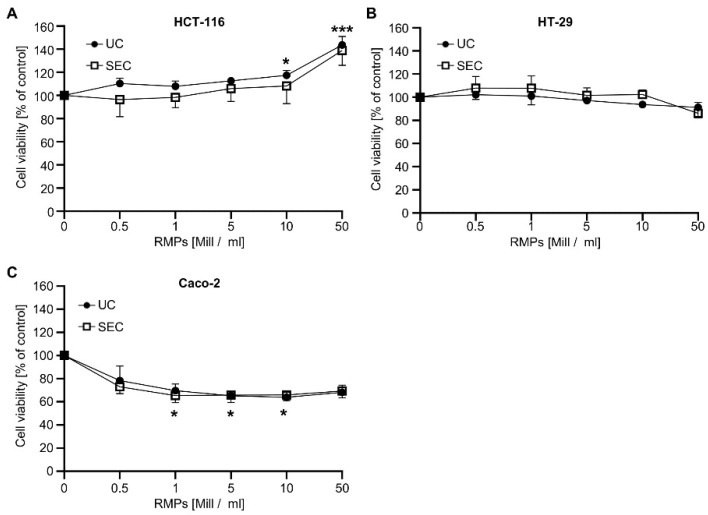
Influence of red-blood cell-derived microparticles on the viability of different colon carcinoma cells.(**A**) HCT-116, (**B**) HT-29 and (**C**) Caco-2 cells were incubated with increasing concentrations of RMPs (0.5–50 × 10^6^ particles/mL) or vehicle control (phosphate buffered saline = PBS). After 48 h, cell viability was assessed by the WST-1 assay. Data represents mean + SD versus vehicle control (100%). At least three independent experiments (each measured in triplicate) were performed for RMPs isolated via ultracentrifugation (UC; *n* = 5) or size exclusion chromatography (SEC; *n* = 3 for HCT-116, *n* = 4 for HT-29 and Caco-2). Statistical analysis: Kruskal–Wallis test followed by Dunn’s multiple comparisons test compared to vehicle control (* *p* ≤ 0.05; *** *p* ≤ 0.001).

**Figure 3 ijms-23-09323-f003:**
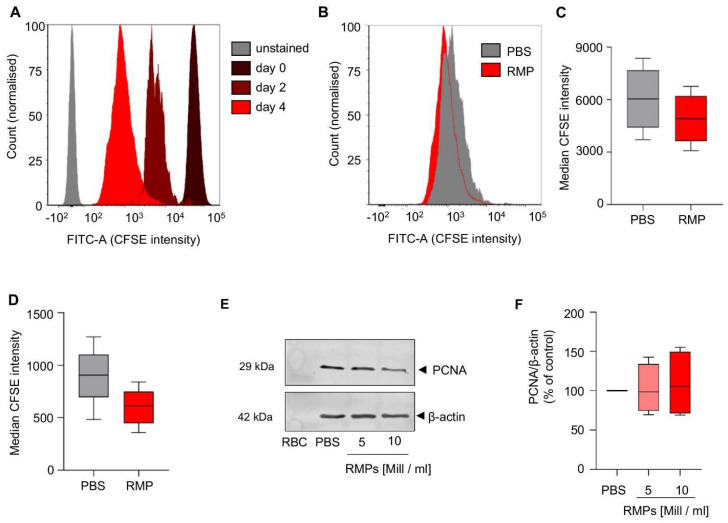
Influence of red blood cell-derived microparticles on the proliferation of HCT-116 cells. (**A**–**D**) HCT116 cells were stained with carboxyfluorescein succinimidyl ester (CFSE) and incubated with 5 × 10^6^ red blood cell-derived microparticles (RMPs)/mL or vehicle control (phosphate buffered saline = PBS). After two and four days, CFSE intensity was measured using flow cytometry (Fluorescein isothiocyanate-A (FITC-A) channel). (**A**) Data shows the intensity of CFSE on day 0, day 2 and day 4 of one representative experiment (RMP treated cells), with unstained cells as negative control. (**B**) Overlay of the intensity of CFSE on day four of one representative experiment comparing RMP-treated cells and vehicle control. (**C**,**D**) Data represents median, 25th to 75th percentile (box), minimum to maximum (whiskers) of RMP treated cells versus vehicle control on day two (**C**) and four (**D**) of five independent experiments (*n* = 5). (**E**,**F**) HCT-116 cells were incubated with the indicated RMP concentrations or vehicle control. After 24 h, cells were lysed and PCNA (proliferating cell nuclear antigen) levels were assessed by Western blot analysis. β-actin served as loading control. Lysed red blood cells (RBCs) were used as negative control. (**E**) Shown is one representative Western Blot out of four independent experiments. (**F**) Quantification of PCNA expression after normalization to loading control (*n* = 4). Data represents median, 25th to 75th percentile (box), minimum to maximum (whiskers) versus vehicle control.

**Figure 4 ijms-23-09323-f004:**
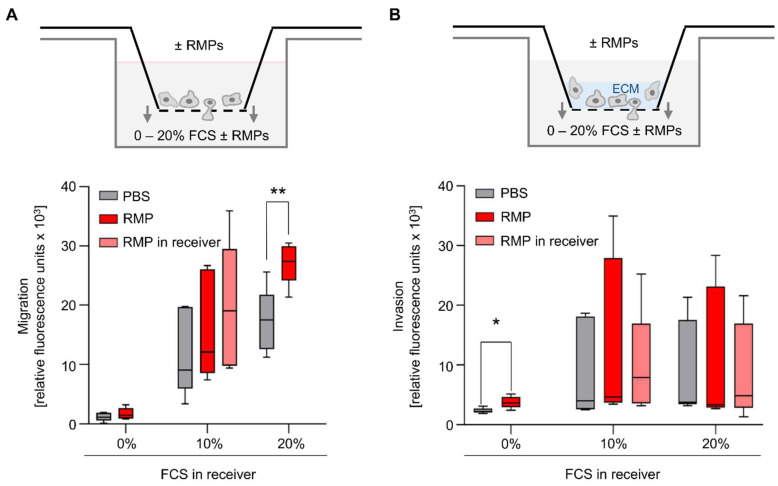
Influence of red blood cell-derived microparticles on migration and invasion of HCT116 cells. (**A**,**B**) HCT116 cells were serum-starved for 24 h and then plated into transwell inserts with a permeable membrane (0.8 µM pore size). Fetal calf serum (FCS) was used as a chemoattractant (see scheme) in the receiver wells. Cells were incubated with 5 × 10^6^ red blood cell-derived microparticles (RMPs)/mL or vehicle control (phosphate buffered saline = PBS), or they were incubated with PBS, while the receivers contained 5 × 10^6^ RMPs/mL added to the FCS. After 24 h, migrated cells were measured using calcein AM. (**B**) To investigate the effects of RMPs on the invasiveness of HCT116 cells, the assay was performed after coating the membranes with extracellular matrix (ECM; Matrigel-Matrix). (**A**,**B**) Data represents median, 25th to 75th percentile (box) and minimum to maximum (whiskers) versus vehicle control. Five independent experiments were done to examine the influence on (**A**) migration (**B**) and invasion. Statistical analysis: unpaired *t*-test (* *p* ≤ 0.05; ** *p* ≤ 0.01).

**Figure 5 ijms-23-09323-f005:**
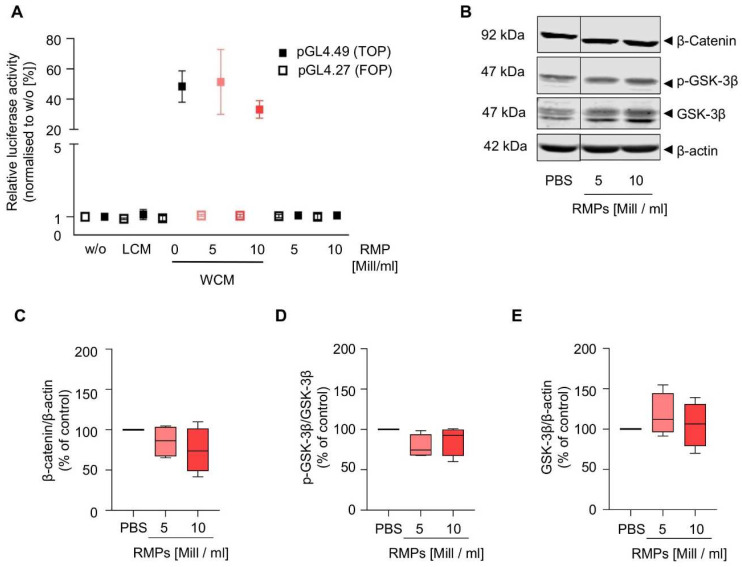
Influence of red blood cell-derived microparticles on the Wnt signaling pathway in HEK293T and HCT-116 cells. (**A**) Wnt reportergene assay. 24 h after co-transfection with either pGL4.27 (TOP) or pGL4.49 (FOP; negative control) with pGL4.74 (Renilla plasmid, transfection control), HEK293T cells were incubated with L-cell conditioned medium (LCM; negative control), Wnt-3A conditioned medium (WCM; containing active Wnt-3A protein), or WCM + one of two concentrations of red blood cell-derived microparticles (RMPs; 5 × 10^6^ or 10 × 10^6^ particles/mL). After 24 h, the Wnt signaling related transcription of the reportergene was assessed via the Dual-Glo^®^ luciferase assay system. Data represents median plus SD after normalization to renilla activity and untreated cells (*w*/*o*). Three independent experiments (each measured in triplicate) were performed. (**B**–**E**) HCT-116 cells were incubated with two concentrations of RMPs (5 × 10^6^ or 10 × 10^6^ particles/mL), vehicle control (PBS). After 24 h, cells were lysed and the protein levels of β-Catenin, p-GSK3β and total GSK3β were assessed by Western blot analysis. β-actin was used as loading control. Lysed red blood cells (RBCs) were used as negative control. (**B**) One representative Western blot is shown out of four independent experiments (*n* = 4) (**C**) Quantification of β-catenin (**C**), p-GSK-3β (**D**) and GSK-3β (**E**) levels after normalization to either loading control (**C**,**E**) or GSK-3β (**D**) Data represent median, 25th to 75th percentile (box), minimum to maximum (whiskers) versus vehicle control (*n* = 4).

**Figure 6 ijms-23-09323-f006:**
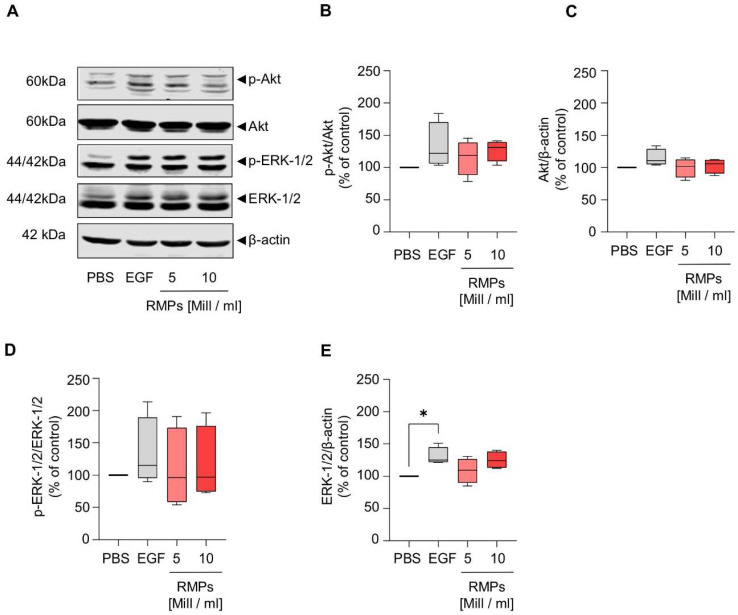
Influence of red-blood cell-derived microparticles on Akt and ERK signaling in HCT-116 cells. HCT-116 cells were incubated with two concentrations of RMPs (5 × 106 or 10 × 106 particles/mL), vehicle control (PBS) and EGF (50 ng/mL) as positive control. After 24 h (RMP and PBS treatment) or 30 min (EGF treatment), whole cell lysates were prepared and the protein levels of p-Akt, Akt, pERK-1/2 and ERK were assessed by Western blot analysis. β-actin was used as loading control. (**A**) One representative Western blot is shown out of four independent experiments (*n* = 4) (**B**–**E**) Quantification of p-Akt (**B**), Akt (**C**), p-ERK-1/2 (**D**) and ERK-1/2 (**E**) levels after normalization to either loading control (**C**,**E**), Akt (**B**) or ERK-1/2 (**D**). Data represents median, 25th to 75th percentile (box), minimum to maximum (whiskers) versus vehicle control (*n* = 4). Statistical analysis: Kruskal–Wallis test followed by Dunn’s multiple comparisons test compared to vehicle control (* *p* ≤ 0.05).

## Data Availability

Not applicable.

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
