# Peer review of "Red Blood Cell-Derived Microparticles Exert No Cancer Promoting Effects on Colorectal Cancer Cells In Vitro"

_ijms, 2022, doi:10.3390/ijms23169323_

Round 1
Reviewer 1 Report
Dear Authors,
The revised version of the manuscript is substantially improved. The results are presented clearly and correctly. Statistical methods are adequately applied.
I have some minor comments:
In my opinion the sentence "These are nanosized extracellular vesicles, which are generated from a broad spectrum of cells and comprise a very heterogeneous group of particles, containing various proteins, lipids, DNA, mRNA, and noncoding RNAs " (line 289 – 291) should be moved in section "Introduction"
The authors mention that they used 36 different RBC units as a source for RMP. If possible could you specify the average age of blood donors?
After these small remarks, I advise on its acceptance in the present form.
Author Response
Thank you very much for the appreciative comments and for the constructive criticism in the past that enabled us to improve our manuscript!
The sentence regarding origin and content of microparticles was moved to the introduction as suggested (line 68).
The average age of the blood donors from whose donations the RMPs used in our project originated was 42 (median 43). We added the information in the results section. Thank you very much for the suggestion.
Reviewer 2 Report
Authors resubmitted their study, and enrich the article to demonstrate that RBC-derived microparticles exert has no cancer promoting effects on colorectal cancer cells in vitro. This study is accepted for publication.
Author Response
We thank the reviewer for the positive evaluation of our resubmitted manuscript, which we were able to improve according to the reviewer's previous constructive criticism.
This manuscript is a resubmission of an earlier submission. The following is a list of the peer review reports and author responses from that submission.
Round 1
Reviewer 1 Report
In this manuscript, Fischer et al. described a study in which they tried to define the observation of why allogeneic transfusion of RBC seems to be linked with poor prognosis of colorectal cancer. The authors hypothesized that the microparticles derived from RBCs might play a specific role in carcinogenic features of colorectal cancer cells, however, the results suggested that the treatment of microparticles showed no significant effects on carcinogenic features in colorectal cancer cells. This is an interesting issue, but this question remains unknown. Some questions are listed below.
- The examination on more colon cancer cell lines is needed to avoid the bias of clone variation, and HEK293T cells are not colon cancer cells.
- In addition to cell proliferation, migration and invasion, the positive effects caused by RMP should be done to demonstrate these RMPs are workable.
- Do the authors check the contents of RMPs from different donors? The variety of packages in different donors’ RMPs may cause other effects.
- Why do the authors just check the impact of RMPs on the Wnt pathway? Other dominant oncogenic pathways in colorectal tumorigenesis should also be tested.
Reviewer 2 Report
Although the previous studies indicated that RBC-derived microparticles (RMPs) are associated with cancer progression, this study firstly demonstrated the relationship between RMPs and cancer progression. However, more evidence is needed to support their study.
Major
- Whether does the source of RMPs affect the experiments? Donors’ Age, gender, or behavior may be involved in regulating RMPs, and there is no effect of RMPs on cancer development. For instance, RMPs from young people or middle-aged people may cause a different effect on cancer progression?
- Thus, dose RMPs components determine cancer progression?
- Dania Fischer et al. utilized HCT-116 cancer cells to demonstrate whether RMPs affected cancer development. Despite no cancer promotion, other colorectal cancer cell lines, such as HT29 or DLD-1 can be applied to support this study.
- Dania Fischer et al. quantified PCNA as a proliferation marker. Cancer cells always keep their proliferation. According to this case, PCNA as a proliferation marker may be difficult to support whether RMPs affect cancer development. Cell viability assay or cell cycle assay is necessary to apply in this study.
- Dania Fischer et al. analyzed GSK-3β in HCT-116 cells, which mediates increased proliferation, migration, and invasion via activation of the Wnt/β-catenin pathway in colorectal carcinoma. RMP incubation affected neither phosphorylation nor the expression of GSK-3β. How about other signal molecules in the Wnt/β-catenin pathway?
- Besides, AKT and MAPK signaling pathways are also involved in regulating cancer development. Dose RMPs affect AKT and MAPK signaling pathways?
Minor
- In Fig. 4, the Figure legend indicated “FBS”, but the figure revealed “FCS”.
- Because “RBC-derived microparticles (RMPs)” was described in the introduction, it is not necessary to mention it again in discussion.
Reviewer 3 Report
Dear Authors,
The idea and the design of the study are interesting and the results are well presented. The methods are described in detail. The Figures are correctly presented. They are of good quality and informative enough. Overall, the study is well performed.
Before the manuscript is published, I have a few recommendations:
In my opinion, the "Discussion" section is too extensive and, in some paragraphs repeats in meaning the text of the “Results” section. Also, the last paragraph should be separated in the section "Conclusions".
There are some typos.
I can recommend the publication after these small corrections will be done.
Round 2
Reviewer 1 Report
In this revised version of the manuscript, the questions have been answered adequately and the title is more appropriate. It has nearly reached the level for publication.
Author Response
We thank the reviewer for the positive statement and the helpful inputs from the first revision. In this new version of the manuscript, we downscaled the title and would suggest: "Red blood cell-derived microparticles do not influence growth, invasion, migration and tumor marker expression of HCT-116 colorectal cancer cells in vitro" in the hope that this will address any last hesitations to publish our manuscript.
Reviewer 2 Report
Although RMPs are very important and novel issue, more evidence is needed to support your study. Please modify your article, and resubmit it again. Actually, “no difference” needs to de carefully demonstrated, otherwise many questions will be coming and fight your study.
Author Response

(The authors gave the same response as above.)

Round 3
Reviewer 2 Report
This article needs more evidence to support authors' results. Please enrich the results and resubmit this article again.